# Helical Hybrid Nanostructure Based on Chiral M13 Bacteriophage via Evaporation-Induced Three-Dimensional Process

**DOI:** 10.3390/nano14141208

**Published:** 2024-07-16

**Authors:** Thanh Mien Nguyen, Sung-Jo Kim, Dae Gon Ryu, Jae Hun Chung, Si-Hak Lee, Sun-Hwi Hwang, Cheol Woong Choi, Jin-Woo Oh

**Affiliations:** 1BK21 FOUR Education and Research Division for Energy Convergence Technology, Pusan National University, Busan 46241, Republic of Korea; ntmien93@pusan.ac.kr; 2Institute of Nanobio Convergence, Pusan National University, Busan 46241, Republic of Korea; sungjokim84@pusan.ac.kr; 3Department of Internal Medicine, Medical Research Institute, Pusan National University School of Medicine and Research Institute for Convergence of Biomedical Science and Technology, Pusan National University Yangsan Hospital, Yangsan 50612, Republic of Korea; gon22gon@naver.com; 4Department of Surgery, Pusan National University School of Medicine and Research Institute for Convergence of Biomedical Science and Technology, Pusan National University Yangsan Hospital, Yangsan 50612, Republic of Korea; jhchung@pnuyh.co.kr (J.H.C.); ghost109@hanmail.net (S.-H.L.); shhwang@pnuyh.co.kr (S.-H.H.); 5Department of Nanoenergy Engineering and Research Center for Energy Convergence Technology, Pusan National University, Busan 46241, Republic of Korea

**Keywords:** chirality, M13 bacteriophage, self-assembly, helical nanostructure, nanoparticle, 3D printing

## Abstract

The use of naturally sourced organic materials with chirality, such as the M13 bacteriophage, holds intriguing implications, especially in the field of nanotechnology. The chirality properties of bacteriophages have been demonstrated through numerous studies, particularly in the analysis of liquid crystal phase transitions, developing specific applications. However, exploring the utilization of the M13 bacteriophage as a template for creating chiral nanostructures for optics and sensor applications comes with significant challenges. In this study, the chirality of the M13 bacteriophage was leveraged as a valuable tool for generating helical hybrid structures by combining it with nanoparticles through an evaporation-induced three-dimensional (3D) printing process. Utilizing on the self-assembly property of the M13 bacteriophage, metal nanoparticles were organized into a helical chain under the influence of the M13 bacteriophage at the meniscus interface. External parameters, including nanoparticle shape, the ratio between the bacteriophage and nanoparticles, and pulling speed, were demonstrated as crucial factors affecting the fabrication of helical nanostructures. This study aimed to explore the potential of chiral nanostructure fabrication by utilizing the chirality of the M13 bacteriophage and manipulating external parameters to control the properties of the resulting hybrid structures.

## 1. Introduction

Chirality is a well-known natural phenomenon that influences the function and origin of organisms at numerous hierarchical levels, including DNA, proteins, peptides, and bacteriophages [1,2]. Inspired by nature, complex chiral metal nanostructures are being rapidly developed using top–down techniques such as electron-beam lithography [3,4], focused ion beam lithography [5], and laser writing [6]. However, fabricating chiral metal nanostructures often requires the creation of three-dimensional (3D) architectures, which is still challenging with top–down fabrication methods [7]. Over the past two decades, researchers have sought to manipulate chiral molecules derived from nature to synthesize and fabricate chiral plasmonic nanoparticles with a more arbitrary structure and abundant biochemical functionalization [8]. Among them, bottom-up techniques such as chemical synthesis on the chiral scaffold, bio-templating [9], DNA origami [10], and liquid crystal dispersion [11,12] have emerged as a versatile platform in the fabrication of chiral nanoplasmonic structures.

Bottom-up approaches are often faster, cheaper, and more versatile in preparing arbitrary structural geometries at a wider scale range. In contrast to the top–down approach, the bottom-up method involves arranging nanoparticles into more complex hierarchical structures, enabling an auspicious way to design unprecedented 3D chiral nanoplasmonic systems exhibiting new physical behavior [8]. This ingenious combination of nanoparticles and chirality molecules has yielded significant results in various domains, including probes, biosensors [13], catalysis [14], and chiral metamaterials [6]. Meanwhile, many structures have been successfully fabricated by the bottom-up method using assemblies, chemical synthesis from DNA, and biopolymer-based materials. Nevertheless, the lack of important controllable properties, such as deposition position on the desired substrate, composition, and dimension of chiral nanoplasmonic structure, make it difficult to meet on-demand fabrication requirements [15]. Three-dimension nanoprinting (3D printing) techniques have been demonstrated as an excellent on-demand fabrication option in broad applications such as photo-emitting materials [16], conductive materials [17,18], and nanophotonic waveguides [19]. Three-dimensional printing at the nanoscale combined with chiral molecules is a solution to overcome the remaining limitations to develop structures with miscellaneous compositions and offering easy control of the dimensions on the desired substrates [20].

On the other hand, a filamentous phage such as M13 is rod-shaped as a biological building block, and it exhibits the lyotropic liquid–crystalline behavior in suspension [21,22,23]. A simple method consisting in drawing substrates vertically from the M13 phage suspension can generate high-order chiral architectures of a hierarchical organization and helical twist [24,25]. By controlling the phage concentration and pulling speed, three unique structures can be achieved: nematic orthogonal twists, cholesteric helical ribbons, and smectic helicoidal nanofilaments. Furthermore, the M13 phage could easily be modified with a genetically engineered technique to create various functionalization properties depending on the specific requirement of each application such as gas sensing [26,27], energy generator [28,29,30], solar cell [31], tissue regeneration [32], and scaffold for a hybrid advanced nanostructure. Due to its unique behavior, the chiral M13 bacteriophage can be an appropriate chiral material combined with metal nanoparticles to generate potential chiral nanoplasmonic structures with both strong optical properties and those used in many biological applications.

In this study, we present a simple technique for 3D nanoprinting to fabricate helical twist nanostructures using a self-assembled M13 phage to guide the arrangement of metal nanoparticles, such as spheres and rods. The study of self-assembled 3D structures, based on the phage as the building block to control the order of the nanoparticles, has been conducted through a home-built 3D printing system with controlled nanopipettes. In this research, parameters affecting the self-assembly process were optimized, including the ratio mixing between M13 phage and nanoparticles and nanopipette diameters. From SEM measurements, the helical structure of the nanoparticles has been successfully observed in various forms. The ratio between nanoparticles and phage has been shown to control the arrangement of nanoparticles in the 3D structure. This design of a 3D helix structure based on the combination of biomaterial and metal nanoparticles can be applied to chiroplasmonic sensors.

## 2. Results and Discussion

### 2.1. M13 Chirality Property and Schematic Illustration of 3D Printing

Figure 1a shows the rod-like structure of the M13 bacteriophage, measuring 880 nm in length and approximately 6.6 nm in diameter. The single-stranded DNA (ssDNA) of the M13 phage is enveloped by thousands of helically arranged pVIII proteins, which contribute to the chirality property of the M13 bacteriophage [24,25]. Due to their helical structure, anisotropic shape, and monodispersion, the M13 phage exhibits a liquid crystalline phase transition based on its concentration. The chirality property of the M13 phage is thoroughly explained by observing the smectic phase transition during the self-assembly process. Consequently, numerous studies have evaluated the liquid crystal (LC) transition process of the M13 phage, progressing from isotropic to nematic, cholesteric, and smectic phases [24,33]. These studies are conducted through various methods using water evaporation-induced processes including pulling-induced methods [24], droplet casting [34], and meniscus dragging deposition (MDD) [35]. Although substantial results have been achieved in understanding the liquid crystalline properties of the M13, applying these characteristics to specific applications remains challenging. In this research, we aim to investigate the arrangement of metal nanoparticles during the self-assembly process of the M13 bacteriophage to create a helical structure based on the chirality properties. This will be achieved through evaporation-induced three-dimensional printing using nanopipettes, as schematically shown in Figure 1b.

Initially, the printing ink solution was prepared by thoroughly mixing various concentrations of nanoparticles in the M13 phage solution. Upon filling the ink solution into a nanopipette, the Si substrate, controlled by the x-y-z stage, approaches and contacts the ink-filled pipette, creating the meniscus at the nanopipette and Si substrate gap. Evaporation and gravity at this interface induce capillary action and convective airflow, transporting M13 bacteriophages and nanoparticles toward the meniscus interface. As the solvent evaporates, the concentration of both M13 bacteriophages and nanoparticles increases. The bacteriophages naturally align and form helices due to their inherent propensity for helical packing. Concurrently, Ag nanoparticles interact with the bacteriophages, influenced by solvent evaporation and capillary forces, aligning and integrating into the helical structures. Upon completion of evaporation, the nanoparticles and bacteriophages are fixed in place, resulting in the formation of stable helical nanostructures on the substrate.

This self-assembly processing is guided by the constant pulling speed (v) (a few µm s^−1^). The termination of the 3D printing process occurs by surpassing the threshold pulling speed, as shown schematically in Appendix A.

### 2.2. Variation in 3D Nanostructures

During the 3D printing process to fabricate pillars, there is a reciprocal influence among several factors, including the concentration of background materials such as M13 bacteriophage, nanoparticles, and pulling speed. Changes in these factors lead to transformations in the structure, contributing to the diversity in the 3D printing architecture. While the inner diameter of the micropipette was kept at approximately 2 µm, the concentration of M13 bacteriophage and nanoparticles was altered to observe the arrangement of the particles in the M13 matrix. Consequently, the arrangement of nanoparticles differs as the nanoparticle concentration increases to a specific virus concentration, leading to diversity in the 3D printing structure, as shown in Figure 2a.

At a low concentration of 0.1 mg/mL, the distribution of nanoparticles is widely spaced, with minimal interaction between nanoparticles due to the small ratio between nanoparticles and M13. As the concentration increases from 0.5 to 1.0 mg/mL, clusters of nanoparticles appear more frequently, randomly distributed within the pillar structure. The influence of M13 during the self-assembly process becomes evident as the concentration rises to 1.5 and 1.7 mg/mL, forming wave-like structures through nanoparticle alignment.

Under high concentration conditions of 2 mg/mL, twisted structures emerge, guided by the M13 bacteriophage during the self-assembly process. The dark-field scattering optical spectra are influenced by the localized surface plasmonic resonance of nanoparticles (Figure 2b). At low concentrations of Ag nanoparticles, the scattering spectra are primarily contributed by single nanoparticles with a resonance wavelength around 500 nm. As the concentration of Ag nanoparticles increases, nanoparticle clusters are formed through the assembly process, leading to strong plasmonic resonance between each nanoparticle in the cluster. Consequently, the scattering shifts significantly to approximately 750 nm. Therefore, the scattering spectrum of 3D nanostructures can be controlled by adjusting the ratio between nanoparticle and M13 bacteriophage concentrations in the printing mixture. Moreover, the hybrid M13 bacteriophage and metal particles can be utilized as optical gas sensors because the M13 bacteriophage is highly sensitive to volatile organic compounds (VOCs) via intermolecular interactions, causing swelling and tuning the gap between nanoparticles in clusters, resulting in changes in the scattering spectral response.

### 2.3. Helical Hybrid Nanostructure with Sphere Shape

To deeply understand the effect between the M13 bacteriophage and nanoparticles, the diameter of the micropipette was reduced to approximately 600 nm. Three-dimensional printing structures are strongly influenced by the ratio between AgNPs and phages. The images display the results of weight ratio (AgNPs/Phage) increasing from left to right corresponding to 1.0, 0.5, 0.3, and 0.1 mg/mL of the M13 phage in fixed 2 mg/mL AgNPs corresponding to the ratios of 2, 4, 6.6, and 20, respectively. The images show the uniform diameter 3D printing structure with approximately 600 nm from the FE-SEM results (Figure 3a). The AgNPs’ density was observed to be higher with an increasing weight ratio. In the first condition (ratio of 2.0), the helical structure was not displayed compared with the other conditions. That was explained by the lack of AgNPs relative to the phage density leading to the unevenly arranged structures, as observed. When the ratio of AgNPs to phage is greater (ratio of 4.0), the number of AgNPs increases significantly, creating a clear helical nanostructure with a helix diameter approximately the same as that of the 3D structure. The AgNPs were highly arranged in a helical organization with sufficient size to form a long chain. At the higher ratios of 6.6 and 20, the AgNPs’ density in the SEM images become denser, especially at the ratio of 20, when a dense helical structure of nanoparticles is formed.

To understand the influence of pulling speed on the self-assembled helical hybrid structure, the pulling speed in the 3D printing process was investigated at three specific values with the same ratio condition of 4.0 (AgNPs/phage). The SEM results are shown in Figure 3.

Two parameters of the helical structure, including pitch length (p) and helical angle (Ψ), were evaluated from the SEM results. The pulling speed was enhanced from the initial condition of 0.15 (µm s^−1^) to 0.3 μm/s, and 0.5 µm s^−1^. In the initial condition, the pitch length (p) and helical angle (Ψ) are of approximately 725 nm and 43°. The ordered shape of the internal arrangement of AgNPs changes significantly when increasing the pulling speed. The increase in pulling speed affects the self-assembly process by altering the dynamics of how nanoparticles and the M13 bacteriophage align. Higher pulling speeds increase the hydrodynamic force acting on the M13 bundles, causing them to stretch more. This elongation effect is similar to the angular control of M13 phage alignment demonstrated in previous studies [25].

Specifically, the helical angle at a higher pulling speed increased to 50° and 57°, which resulted in the pitch (p) also gradually becoming greater, from 715 nm (0.15 µm s^−1^) to 877 nm at 0.3 (µm s^−1^) and 1122 nm at (0.5 µm s^−1^), respectively. The SEM results can conclude that the pulling speed can affect the self-assembly processing in 3D printing by changing the pitch and helical angle value. Consequently, the optical property of the helical hybrid structure can vary under parameter control following our desired structure.

### 2.4. Helical Hybrid Nanostructure with Rod Shape

The investigation into the diversity of particles has garnered significant attention, particularly in applications involving chiral plasmonic materials based on helical structures. Among the various chiral plasmonic materials, gold nanorods (GNRs) have attracted considerable interest due to their biocompatibility, ease of surface functionalization, large gyromagnetic ratios (g-values), and near-infrared absorption capabilities. Furthermore, owing to their anisotropic shape and their ability to adjust the aspect ratio, they exhibit chiroptical properties, making them suitable for applications in chiral plasmonics. In a previous study, the combination of gold nanorods with the M13 bacteriophage was explored using the MDD method [25]. In the MDD process, a liquid meniscus forms between a moving solid substrate and a blade surface. As the solid substrate moves, a thin film comprising the M13 bacteriophage and GNRs is deposited on the substrate.

This exploration has successfully demonstrated that the GNRs can be assembled by the evaporation-induced self-assembly of the M13 bacteriophage that revealed promising results for generating a helical structure in the 3D printing process. To achieve helical nanostructure assembly through the 3D printing process, various concentrations of Au nanorods were mixed with the M13 bacteriophage at a concentration of 0.3 mg/mL (Figure 4a). The gold nanorods, with lengths and diameters of 55 nm and 15 nm, respectively, and an aspect ratio of 4.1, were incorporated into the mixed solution. This solution was then injected into a nanopipette for the 3D printing process at a rate of 0.15 µm s^−1^ on a silicon substrate, followed by the evaluation of the dried 3D pillar nanostructure of the M13–GNRs composite through SEM. SEM images illustrate how the GNRs are helically well-arranged particles within the M13 matrix at various concentrations. At lower concentrations, discrete GNRs were observed, and the chiral property of the M13 bacteriophage during self-assembly could not be discerned (Figure 4b). As the GNR concentration increased, the alignment between the GNRs became more pronounced, as confirmed by SEM observations (Figure 4c–e). At higher concentrations of GNRs (0.8–1.2 mg/mL), the GNRs approached and aggregated in a shoulder-to-shoulder manner. In this scenario, the chirality property of the M13 bacteriophage is evident based on the GNRs’ arrangement in a 3D pillar structure. Notably, further increasing the GNR concentration facilitates the fabrication of the helical structure.

The alignment of these nanorods in the M13 matrix is attributed to the self-assembled M13 bacteriophage in 3D printing. Moreover, the properties of the M13 bacteriophage can be tailored through genetic engineering techniques to manifest various controllable characteristics, presenting an attractive feature for the fabrication of diverse nanostructures. This paves the way for the utilization of orientation-controlled M13 bacteriophage as templating tools for guest materials such as GNRs, thereby inducing chirality and potentially enhancing optical properties.

Overall, our approach has successfully fabricated a helical hybrid nanostructure by combining the chiral M13 bacteriophage with metal nanoparticles. Our system offers several advantages, including the utilization of 3D printing techniques and the biomaterial properties of the M13 bacteriophage.

Firstly, the investigation into the ratio effect between the M13 bacteriophage and nanoparticles enables the creation of various 3D nanostructures with unique shapes, including clusters, waves, and hierarchical organizations such as helical structures. This versatility in nanostructure design opens up possibilities for tailored applications across different fields. Secondly, the exploration of pulling speed in the 3D printing process demonstrates its impact on the arrangement of structures in terms of pitch length and helical angle. By leveraging 3D printing techniques, the researchers showcase a scalable and customizable approach to fabricating chiral nanoplasmonic structures. Thirdly, the M13 bacteriophage is sensitive to environmental stimuli such as pH, temperature, and the presence of specific molecules. This sensitivity can be harnessed for sensing applications, wherein changes in structure or behavior in response to environmental cues can be detected and quantified. However, some limitations remain to be addressed, including the need for further investigation into the fabrication of helical structures with controlled left or right-handedness. This can be achieved by using a genetically engineered M13 bacteriophage.

## 3. Conclusions

We successfully fabricated a helical hybrid nanostructure by combining the chiral M13 bacteriophage with nanoparticles. High-order arrangements of nanoparticles in both sphere and rod shapes were observed through SEM measurements. An investigation into the ratio effect between the M13 bacteriophage and nanoparticles was conducted, leading to the creation of various 3D nanostructures, each possessing a unique shape ranging from clusters and waves to high hierarchical organizations, such as a helical structure. Consequently, the optical properties of each unique structure were evaluated using dark-field scattering. Furthermore, by exploring the pulling speed of the 3D printing process, we demonstrated its effect on the arrangement of structures in pitch length and helical angle. With the advantages of a 3D structure, including controllable components and dimensions, the helical hybrid nanostructure achieved through 3D printing techniques presents a promising approach for fabricating chiral nanoplasmonic structures that can be used in various applications. In optoelectronics, the precise control over the arrangement of nanoparticles can lead to improved performance in devices such as photodetectors and light-emitting diodes. In sensing technologies, wherein the M13 bacteriophage can be utilized as a receptor for VOC detection, the unique optical properties of these nanostructures can be leveraged for the highly sensitive detection of biological and chemical substances.

## 4. Experimental Section

### 4.1. Preparation of the M13 Bacteriophage Solution

The M13 bacteriophages were prepared following a standard mass amplification protocol. After amplification, the bacteriophages underwent purification and concentration through polyethylene glycol precipitation. A suspension of bacteriophages was then prepared by dispersing them in a tris-buffered saline solution with a pH of 7.5, containing 12.5 mM tris and 37.5 mM NaCl. The concentration of the suspension was adjusted and determined by measuring the optical absorbance using a UV–visible spectrophotometer (NanoDrop-One Microvolume, Thermo Fisher Scientific, Waltham, MA, USA).

### 4.2. Three-Dimensional Printing of Helical Nanostructure

The 3D printing process is conducted using a home-built experiment, as shown schematically in Appendix A. The optical system includes a halogen fiber illuminator (OSL2, Thorlabs, Newton, USA), objective lens (20x) (MY20X-804, Mitutoyo, Kanagawa, Japan), and charge-coupled device (CCD) camera was prepared to observe the printing process. The pulling process is generated by moving of x-y-z stage (M-VP-25XA-XYZR, Newport, Irvine, USA) with the Si substrate on the top. All the set up of the 3D printing system is shown in Appendix A.

The 3D printing is conducted using a nanopipette with a 600 nm inner diameter (as shown in Appendix A). The diameter of the nanopipette was obtained by using the typical glass capillary puller (P-97, Sutter Instruments, Novato, USA). The heating time and pulling velocity were optimized to achieve the desired diameter. After injecting ink (mixing of phage and nanoparticles) into the nanopipette, all 3D printing processes, including approach, contact, guide, and termination, were controlled by the x-y-z stage via the controller.

The ink solution in this study is prepared by mixing M13 bacteriophage and Ag nanoparticles (75 nm) or gold nanorod with length and diameter of 55 nm and 15 nm (Nanocomposite, San Diego, USA), as shown detailed in TEM images (Appendix A). Before mixing with M13 bacteriophage, nanoparticles are centrifuged to remove the surfactant in the origin solution, and then dispersed in water. The hybrid solution of nanoparticle and phage was vortexed to well-dispersion.

### 4.3. Characterization of 3D Pillar Structure

The structure characterization of the 3D structure was measured by scanning electron microscopy (SEM) using ultra-high-resolution low voltage FE-SEM (JSM-7900F, PNU center for Research facilities, Busan, Republic of Korea).

### 4.4. Optical Measurements

The scattering spectra of each prepared specimen underwent measurement utilizing a commercial Olympus bright-field/dark-field (DF) microscope (BX53M) equipped with a 100 × 0.9 numerical aperture (NA) objective. This analysis was conducted under unpolarized halogen-light illumination. Furthermore, the scattering spectra were assessed using a fiber-optic spectrometer (US/USB4000, Ocean Optics, Dunedin, FL, USA).

## Figures and Tables

**Figure 1 nanomaterials-14-01208-f001:**
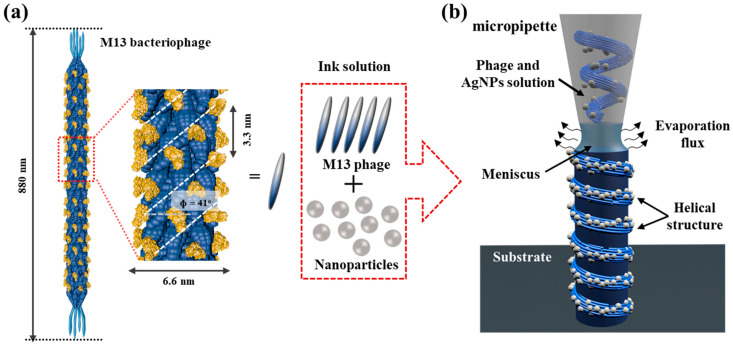
(**a**) Schematic structure of chirality M13 bacteriophage, and the illustration of 3D printing process (**b**).

**Figure 2 nanomaterials-14-01208-f002:**
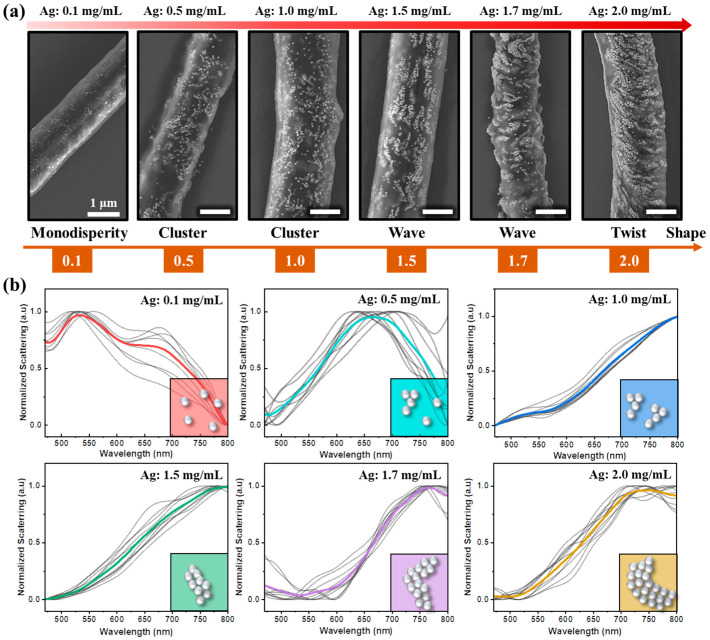
(**a**) SEM images of the 3D pillar structure were obtained by varying the concentration of Ag nanoparticles (0.1, 0.5, 1.0, 1.5, 1.7, 2.0 mg/mL, respectively). (**b**) Scattering resonance spectra were measured from the 3D pillar structures at different concentrations of Ag nanoparticles as shown in (**a**). The inset illustrates the shape of Ag nanoparticles in the 3D pillar structures.

**Figure 3 nanomaterials-14-01208-f003:**
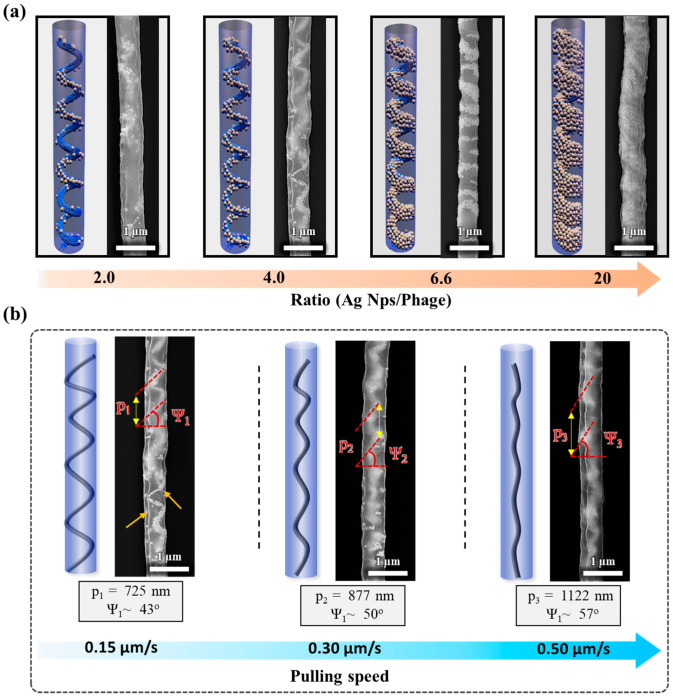
(**a**) SEM images depict the helical hybrid nanostructure composed of AgNPs and bacteriophage at varying weight ratios achieved by adjusting the phage concentration (1.0, 0.5, 0.3, and 0.1 mg/mL, respectively) in a solution containing 2 mg/mL AgNPs. (**b**) Pulling speed dependency on the helical hybrid nanostructure.

**Figure 4 nanomaterials-14-01208-f004:**
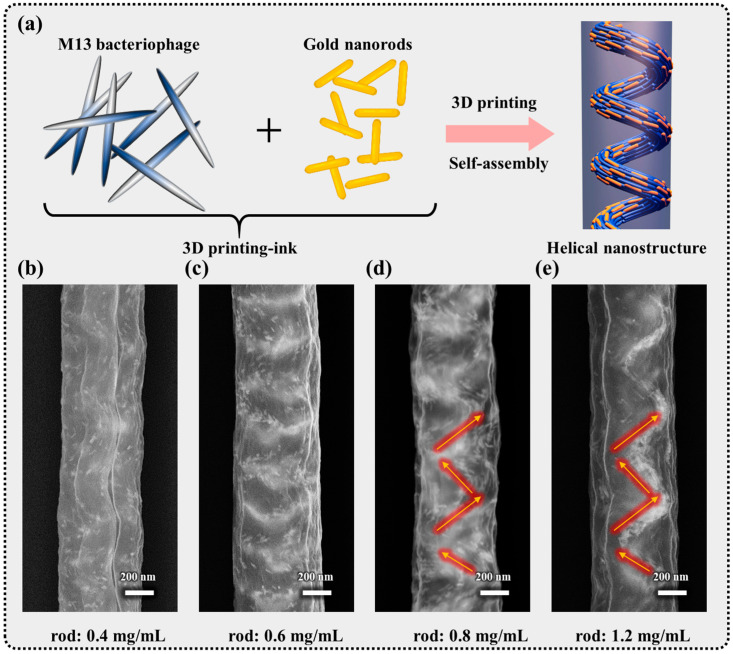
(**a**) A schematic of the 3D printing-ink preparation based on M13 bacteriophage and gold nanorods (GNRs). M13 bacteriophage is used as a scaffold for gold nanorods. During the evaporation of water during 3D printing process, the M13 bacteriophage produces the helical-twisted bundles leading to the arrangement of GNRs in the M13 matrix. (**b**–**e**) SEM images of 3D pillars at various GNR concentrations in the 3D printing-ink from 0.4 mg/mL (**b**), 0.6 mg/mL (**c**), 0.8 mg/mL (**d**), and 1.2 mg/mL (**e**).

## Data Availability

All data generated or analyzed during this study are included in this published article and its and its Appendix A.

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
