# Peer review of "Helical Hybrid Nanostructure Based on Chiral M13 Bacteriophage via Evaporation-Induced Three-Dimensional Process"

_nanomaterials, 2024, doi:10.3390/nano14141208_

Round 1
Reviewer 1 Report
Comments and Suggestions for Authors
In this paper, the authors utilized M13 bacteriophagethe as naturally sourced templates with inherent chirality to fabricate helical hybrid materials by integrating it with nanoparticles through an evaporation-induced 3D printing process. The self-assembly of M13 bacteriophage effectively organize metal nanoparticles into a helical chain at
the meniscus interface. Several parameters influencing the structural characteristics have been discussed in this study. Although the results are interesting, the current manuscript lacks sufficient evidence to fully demonstrate its significance. A major revision is necessary before considering its publication in Nanomaterials. The following are the questions and suggestions about this manuscript:
1. The schematic diagram in Figure 1 lacks clarity and requires a more precise elucidation of the pattern definitions. Additionally, a more detailed explanation is needed to clarify how evaporation induces the packing of M13 bacteriophage and nanoparticles.
2. Why were irregular concentration intervals chosen for the experiments in Figure 2? Have any quantitative conclusions been derived from these results? It is recommended to include an experiment without nanoparticles as a control group. Can this method produce any helical structure using M13 bacteriophage alone?
3. Is the helical structure left-handed or right-handed? Can it be manipulated or customized as needed?
4. What are the distinctions between the experiments pertaining to Ag concentrations or Ag Nps/Phage ratios?
5. The mechanism of pulling speed influence the pitch length and helical angle should be further discussed.
6. What is the driving force for such a helical self-assembly? In terms of the M13 bacteriophage, which factor holds greater significance: geometry or surface chemistry?
7. How do the size and geometry of nanoparticles impact the process of assembly?
8. Silver nanoparticles and gold nanoparticles exhibit antimicrobial properties. Have you investigated the survival rate of M13 bacteriophage in your study?
9. The material structures are intriguing. I believe that demonstrating the practical application of such materials will enhance the appeal of this work to a wide range of audiences.
Comments on the Quality of English LanguageThe English language should be carefully reviewed for any typographical errors and grammatical mistakes.
Author Response
Reviewer 1
Comments 1:
The schematic diagram in Figure 1 lacks clarity and requires a more precise elucidation of the pattern definitions. Additionally, a more detailed explanation is needed to clarify how evaporation induces the packing of M13 bacteriophage and nanoparticles.
Response 1:
Thank you for your valuable comments. The schematic diagram in Figure 1 has been modified to enhance clarity, as shown in the figure below. Additionally, an explanation of how evaporation induces the packing of M13 bacteriophage and nanoparticles has been added at the appropriate part.
+ The self-assembly induced by evaporation during 3D printing can briefly explained as follows:
Due to the chirality property of the M13 bacteriophage, a chiral spiral structure can form through the self-assembly process involving M13 and metal nanoparticles. The chirality of the M13 bacteriophage stems from its intrinsic helical protein coat structure. The self-assembly primarily occurs at the meniscus interface. During the 3D printing process, as the micropipette approaches the Si-substrate, a meniscus interface is created where the ink containing M13 bacteriophages and nanoparticles interacts. Evaporation and gravity at this interface induce capillary action and convective airflow, transporting M13 bacteriophages and nanoparticles towards the meniscus interface. As solvent evaporates, the concentration of both M13 bacteriophages and nanoparticles increases. This reduction in the liquid phase volume brings the components closer together. In systems with dissimilar shapes and sizes—such as rod-shaped M13 bacteriophages and spherical Ag nanoparticles—phase separation, as described by Onsager’s theory (1949), may occur. This process can lead to a rod-rich phase enriched with bacteriophages and a sphere-rich phase enriched with nanoparticles at the meniscus interface.
As evaporation progresses, the bacteriophages naturally align and form helices due to their inherent propensity for helical packing. Concurrently, Ag nanoparticles interact with the bacteriophages, influenced by solvent evaporation and capillary forces, aligning and integrating into the helical structures. Upon completion of evaporation, the nanoparticles and bacteriophages are fixed in place, resulting in the formation of stable helical nanostructures on the substrate.
Added text (Line: 120-131): Initially, the printing ink solution was prepared by thoroughly mixing various concentrations of nanoparticles in the M13 phage solution. Upon filling the ink solution into a nanopipette, the Si substrate controlled by the x-y-z stage, approaches and contacts the ink-filled pipette, creating the meniscus at the nanopipette and Si substrate gap. The high surface-to-volume ratio of the meniscus leads to a rapid evaporation rate of water. Evaporation and gravity at this interface induce capillary action and convective airflow, transporting M13 bacteriophages and nanoparticles towards the meniscus interface. As solvent evaporates, the concentration of both M13 bacteriophages and nanoparticles increases. The bacteriophages naturally align and form helices due to their inherent propensity for helical packing. Concurrently, Ag nanoparticles interact with the bacteriophages, influenced by solvent evaporation and capillary forces, aligning and integrating into the helical structures. Upon completion of evaporation, the nanoparticles and bacteriophages are fixed in place, resulting in the formation of stable helical nanostructures on the substrate.
Consequently, the concentration of M13 bacteriophage increases at the meniscus interface, initiating the self-assembly of M13 bacteriophage and nanoparticles. This self-assembly processing is guided by the constant pulling speed (v) (a few µm s-1). The termination of the 3D printing process occurs by surpassing the threshold pulling speed, as shown schematically in Figure S2b.
Comments 2:
+ Why were irregular concentration intervals chosen for the experiments in Figure 2?
Response 2: Thank you for your question. Initially, the concentrations were optimized according to regular concentration intervals. However, we noticed a significant transition from 1.5 mg/ml to 2 mg/ml. Therefore, to clearly understand the transition process, a concentration of 1.7 mg/ml was added to clarify the structural changes as the concentration of Ag nanoparticles increases.
+ Have any quantitative conclusions been derived from these results? It is recommended to include an experiment without nanoparticles as a control group. Can this method produce any helical structure using M13 bacteriophage alone?
Response 2: Thank you for your value comments. As explained in Figure 1a, because the geometry and surface chemistry of M13 bacteriophage have a helical structure, helical structures appear during the self-assembly process. When M13 bacteriophage is mixed with gold nanoparticles at a very low concentration, the depletion force generated by spherical nanoparticles can be ignored, but when the concentration increases, gold nanoparticles form nanoclusters among themselves.
Comments 3:
Is the helical structure left-handed or right-handed? Can it be manipulated or customized as needed?
Response 3: Thank you for your comments. It has been reported that M13 bacteriophages exhibit chiral structures through self-assembly and that they all have a helicity of ±1 [1]. This means that either left- or right-handed helical structures can be created through 3D printing. However, it is currently difficult to accurately control the helicity of the structure.
Reference:
[1] Chung, W.-J.; Oh, J.-W.; Kwak, K.; Lee, B. Y.; Meyer, J.; Wang, E.; Hexemer, A.; Lee, S.-W., Biomimetic self-templating 370 supramolecular structures. Nature 2011, 478 (7369), 364-368.
Comments 4:
What are the distinctions between the experiments pertaining to Ag concentrations or Ag Nps/Phage ratios?
Response 4: Thank you for your comment. Two parameters were used to describe the same condition: Ag concentrations and Ag NPs/Phage ratios. The Ag concentration refers to the concentration of Ag NPs in the 3D printing ink after mixing with the M13 bacteriophage. To better illustrate the difference between the number of nanoparticles and the number of M13 bacteriophages, a ratio of Ag NPs/Phage is provided based on these concentrations. Even if the Ag nanoparticle concentration remains the same, the ratio will change if the M13 concentration in the ink changes. This ratio helps provide a clearer understanding of the relationship between the number of nanoparticles and the amount of M13 bacteriophage.
Comments 5:
The mechanism of pulling speed influence the pitch length and helical angle should be further discussed.
Response 5:
Thank you for your valuable comment. As you know, the changes in helical structures primarily result from the structural modifications of the M13 bacteriophage during the self-assembly process. All of the self-templated structures are tunable by varying parameters that affect the kinetics and thermodynamics of assembly, such as phage concentration, pulling speed, and phage surface chemistry. Specifically, the effect of pulling speed has been investigated in various reports [1,2,3].
Increased pulling speed causes these bundles to extend due to the increased hydrodynamic force during the formation of the helicoidal liquid crystal phase [2]. Additionally, M13 bundles in solution are much stretched due to the increased capillary flow effect at higher pulling speeds [2]. Therefore, the pitch length and angle of the helical nanostructure can be enhanced because higher speeds increase the hydrodynamic force, which deforms the M13 bundles, causing them to extend more. The authors added an explanation about the pulling speed effect at Figure 3.
Added text [line: 198-208]:
Two parameters of helical, including pitch length (p) and helical angle (Ψ), were evaluated from SEM results. The pulling speed was enhanced from initial condition of 0.15 (µm s−1) to 0.3 μm/s, and 0.5 µm s−1. In the initial condition, the pitch length (p) and helical angle (Ψ) are approximately 725 nm and 43o. The ordered shape of the internal arrangement of AgNPs has changed significantly when increasing the pulling speed. That is reminiscent of the angular control of M13 phage alignment demonstrated by varying the pulling speed in the previous study. The increase in pulling speed affects the self-assembly process by altering the dynamics of how nanoparticles and M13 bacteriophage align. Higher pulling speeds increase the hydrodynamic force acting on the M13 bundles, causing them to stretch more. This elongation effect is similar to the angular control of M13 phage alignment demonstrated in previous studies [2].
References:
[1] Chung, W.-J.; Oh, J.-W.; Kwak, K.; Lee, B. Y.; Meyer, J.; Wang, E.; Hexemer, A.; Lee, S.-W., Biomimetic self-templating supramolecular structures. Nature 2011, 478 (7369), 364-368.
[2] Park, S. M.; Kim, W.-G.; Kim, J.; Choi, E.-J.; Kim, H.; Oh, J.-W.; Yoon, D. K., Fabrication of Chiral M13 Bacteriophage Film by Evaporation-Induced Self-Assembly. Small 2021, 17 (26), 2008097.
[3] Oh, J.-W.; Chung, W.-J.; Heo, K.; Jin, H.-E.; Lee, B. Y.; Wang, E.; Zueger, C.; Wong, W.; Meyer, J.; Kim, C., Biomimetic virus-based colourimetric sensors. Nat. Commun. 2014, 5, 3043.
Comments 6:
What is the driving force for such a helical self-assembly? In terms of the M13 bacteriophage, which factor holds greater significance: geometry or surface chemistry?
Response 6:
The driving force for the helical self-assembly of M13 bacteriophage with nanoparticles is primarily a combination of geometric and surface chemistry factors. The M13 bacteriophage itself is a chiral filamentous virus that naturally tends to form helical structures.
The chirality of the M13 bacteriophage is an intrinsic property derived from its helical structure, created by approximately 2700 copies of the protein coat (pVIII) arranged around the DNA core. The protein coat presents functional groups (such as amine, carboxyl, and thiol groups) on the surface of the bacteriophage. Therefore, the combination of the long shape of M13 bacteriophage and the helical structure of its protein coat enables it to exhibit chirality during the self-assembly process. Thus, both geometry and surface chemistry are critical for the helical self-assembly of the M13 bacteriophage
Comments 7:
How do the size and geometry of nanoparticles impact the process of assembly?
Response 7:
Thank you for your value comment. When gold nanoparticles and M13 bacteriophages are mixed, geometry has the greatest influence on assembly. When gold nanoparticles are spherical, a larger excluded volume occurs compared to when they are rod-shaped. Therefore, spherical gold nanoparticles agglomerate more easily, and when in the rod shape, they tend to mix well parallel to the direction of M13 bacteriophage alignment.
Comments 8:
Silver nanoparticles and gold nanoparticles exhibit antimicrobial properties. Have you investigated the survival rate of M13 bacteriophage in your study?
Response 8:
Thank you for your comment. Both silver (Ag) and gold (Au) nanoparticles indeed exhibit antimicrobial properties. However, in our study, the M13 bacteriophage was used as a scaffold for the self-assembly process, meaning we utilized the structure of the M13 bacteriophage as a building block for the assembly. Therefore, the biological activity of the bacteriophage was not a concern, as all M13 bacteriophages used in our experiments were inactivated.
Comments 9:
The material structures are intriguing. I believe that demonstrating the practical application of such materials will enhance the appeal of this work to a wide range of audiences.
Response 9:
Thank you for your valuable comment. An explanation about the potential applications has been added to the conclusion.
Added text (Line 293-298): In optoelectronics, the precise control over the arrangement of nanoparticles can lead to improved performance in devices such as photodetectors and light-emitting diodes. In sensing technologies, where the M13 bacteriophage can be utilized as a receptor for VOC detection, the unique optical properties of these nanostructures can be leveraged for highly sensitive detection of biological and chemical substances.

Reviewer 2 Report
Comments and Suggestions for Authors
The authors report the fabrication of a helical hybrid nanostructure by combining chiral M13 bacteriophage with Ag nanoparticles by using an evaporation-induced 3D printing process. The helical alignment was confirmed by SEM measurements. They demonstrate that the fabrication of 3D nanostructures is affected by external parameter, such as nanoparticle shape, nanoparticles–M13 ratio, and pulling speed. Although the findings are interesting, the experimental results, i.e., the observed effects of these parameters on the fabrication, should be explained clearly. Furthermore, a phrase of “the driving force of M13 bacteriophage” has been often used in the text. The authors should explain the meaning of this phrase.
Author Response
Reviewer 3
Comments 1:
Although the findings are interesting, the experimental results, i.e., the observed effects of these parameters on the fabrication, should be explained clearly.
Response 1: Thank you for your comment. Following your suggestion, observed effects of these parameters such as pulling speed, concentration was modified clearly.
Revised text:
Added text (Line: 120-131): Initially, the printing ink solution was prepared by thoroughly mixing various concentrations of nanoparticles in the M13 phage solution. Upon filling the ink solution into a nanopipette, the Si substrate controlled by the x-y-z stage, approaches and contacts the ink-filled pipette, creating the meniscus at the nanopipette and Si substrate gap. The high surface-to-volume ratio of the meniscus leads to a rapid evaporation rate of water. Evaporation and gravity at this interface induce capillary action and convective airflow, transporting M13 bacteriophages and nanoparticles towards the meniscus interface. As solvent evaporates, the concentration of both M13 bacteriophages and nanoparticles increases. The bacteriophages naturally align and form helices due to their inherent propensity for helical packing. Concurrently, Ag nanoparticles interact with the bacteriophages, influenced by solvent evaporation and capillary forces, aligning and integrating into the helical structures. Upon completion of evaporation, the nanoparticles and bacteriophages are fixed in place, resulting in the formation of stable helical nanostructures on the substrate.
Consequently, the concentration of M13 bacteriophage increases at the meniscus interface, initiating the self-assembly of M13 bacteriophage and nanoparticles. This self-assembly processing is guided by the constant pulling speed (v) (a few µm s-1). The termination of the 3D printing process occurs by surpassing the threshold pulling speed, as shown schematically in Figure S2b.
Added text [line: 198-208]:
Two parameters of helical, including pitch length (p) and helical angle (Ψ), were evaluated from SEM results. The pulling speed was enhanced from initial condition of 0.15 (µm s−1) to 0.3 μm/s, and 0.5 µm s−1. In the initial condition, the pitch length (p) and helical angle (Ψ) are approximately 725 nm and 43o. The ordered shape of the internal arrangement of AgNPs has changed significantly when increasing the pulling speed. That is reminiscent of the angular control of M13 phage alignment demonstrated by varying the pulling speed in the previous study. The increase in pulling speed affects the self-assembly process by altering the dynamics of how nanoparticles and M13 bacteriophage align. Higher pulling speeds increase the hydrodynamic force acting on the M13 bundles, causing them to stretch more. This elongation effect is similar to the angular control of M13 phage alignment demonstrated in previous studies [2].
Comments 2:
Furthermore, a phrase of “the driving force of M13 bacteriophage” has been often used in the text. The authors should explain the meaning of this phrase.
Response 2:
Thanks for your meticulousness. The authors have edited the phrase "Driving Force of M13 Bacteriophage" to make the exact meaning easier to understand.
Revised text:
Line 179: At the higher ratio of 6.6 and 20, the AgNPs density in SEM images becomes denser, especially at the ratio of 20, a dense helical structure of nanoparticles is formed. under the driving force of M13 bacteriophage.
Line 254: The alignment of these nanorods in the M13 matrix is attributed to the driving force of the M13 bacteriophage during the self-assembled M13 bacteriophage process in 3D printing.
